# ON THE RELATIONSHIP BETWEEN ADVERSARIAL ROBUSTNESS AND DECISION REGION IN DEEP NEURAL NETWORKS

## ABSTRACT

In general, Deep Neural Networks (DNNs) are evaluated by the generalization performance measured on unseen data excluded from the training phase. Along with the development of DNNs, the generalization performance converges to the state-of-the-art and it becomes difficult to evaluate DNNs solely based on this metric. The robustness against adversarial attack has been used as an additional metric to evaluate DNNs by measuring their vulnerability. However, few studies have been performed to analyze the adversarial robustness in terms of the geometry in DNNs. In this work, we perform an empirical study to analyze the internal properties of DNNs that affect model robustness under adversarial attacks. In particular, we propose the novel concept *Populated Region Set (PRS)*, where training samples are populated more frequently, to represent the internal properties of DNNs in a practical setting. From systematic experiments with the proposed concept, we provide empirical evidence to validate that a low PRS ratio has a strong relationship with the adversarial robustness of DNNs. We also devise PRS regularizer leveraging the characteristics of PRS to improve the adversarial robustness without adversarial training.

## 1 INTRODUCTION

With the steep improvement of the performance of Deep Neural Networks (DNNs), their applications are expanding to the real world, such as autonomous driving and healthcare (Huang & Chen, 2020; LeCun et al., 2015; Miotto et al., 2018). For real world application, it may be necessary to choose the best model among the candidates. Traditionally, the generalization performance which measures the objective score on the test dataset excluded in the training phase, is used to evaluate the models (Bishop, 2006). However, it is non-trivial to evaluate DNNs based on this single metric. For example, if two networks with the same structure have the similar test accuracy, it is ambiguous which is better. Robustness against adversarial attacks, measure of the vulnerability, can be an alternative to evaluate DNNs (Szegedy et al., 2013; Huang et al., 2015; Jakubovitz & Giryes, 2018; Yuan et al., 2019; Zhong et al., 2021). Most previous works were focused on the way to find adversarial samples by utilizing the model properties such as gradients with respect to the loss function. Given that the adversarial attack seeks to find the perturbation path on the model prediction surface over the input space, robustness can be expressed in terms of the geometry of the model. However, few studies have been performed to interpret the robustness with the concept of the geometric properties of DNNs. From a geometric viewpoint, the internal properties of DNNs are represented by the boundaries and the regions (Baughman & Liu, 2014). It is shown that the DNNs with piece-wise linear activation layers are composed of many linear regions, and the maximal number of these regions is mathematically related to the expressivity of DNNs (Montúfar et al., 2014; Xiong et al., 2020). As these approaches only provide the upper bound for the expressivity with the same structured model, it does not explain how much information the model actually expresses.

In this work, we investigate the relationship between the internal properties of DNNs and the robustness. In particular, our approach analyzes the internal characteristics from the perspective of the decision boundary (DB) and the decision region (DR), which are basic components of DNNs (Fawzi et al., 2017). To avoid insensitivity of the maximal number of linear regions in the same structure assumption, we propose the novel concept *Populated Region Set (PRS)*, which is a set of

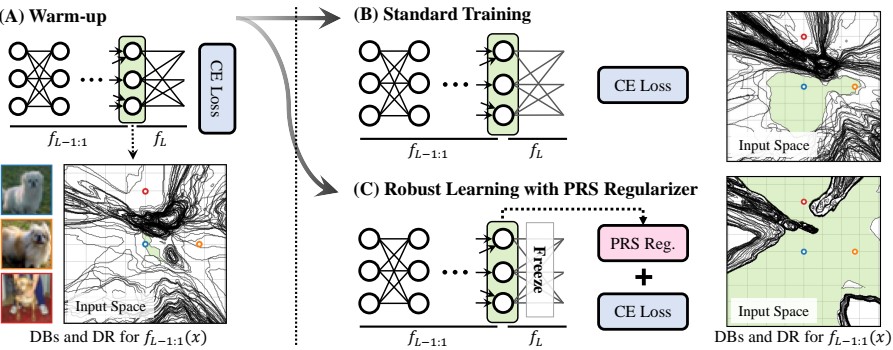

Figure 1: An illustrative comparison of each training method with CIFAR-10, and visualization for decision boundaries/regions (DBs/DRs) of penultimate layer in the input space ($f_{L-1:1}(x)$). For visualization, We randomly select three training images in *dog* class and make a hyperplane with these images. The green area indicates DR which the blue boxed image populates. (A) Warm-up stage for VGG-16 with standard training. (B) Standard training after warm up stage. (C) The robust learning with devised PRS regularizer after warm up stage. We identify each training method induces different configuration of DBs/DRs, which represents different internal properties of DNNs.

DRs containing at least one training sample included in the training dataset. Since the PRS can be considered as the feasible complexity of the model, we hypothesize the size of PRS is related to the robustness of network. To validate our hypothesis, we perform systematic experiments with various structures of DNNs and datasets. Our observations are summarized as follows:

- The models with the same structure can have different size of PRS, although they have similar generalization performance. We empirically show that the model with a small size of PRS tends to show higher robustness compared to that with a large size. (in Section 3.2)

- We observe that when the model achieves a small size of PRS, the linear classifier which maps the penultimate features to the logits has high cosine similarity between parameters corresponding to each class (in Section 3.2).

- We verify that the size of intersection of the PRS from the training/test dataset is related to the robustness of model. The model with a large intersection of training/test dataset has higher robustness than the model with a small intersection (in Section 3.3).

- We devise a novel regularizer leveraging the characteristics of PRS to improve the robust accuracy without adversarial training (in Section 4).

## 2 INTERNAL PROPERTIES OF DNNS

This section describes the internal properties of DNNs in the perspective of DBs and DRs. To expand the notion of DBs and DRs to the internal feature-level, we re-define the DBs in the classifier that generalizes the existing definition of DBs. Finally, we propose *Populated Region Set (PRS)* which describes the specific DRs related to the training samples.

### 2.1 DECISION BOUNDARY AND REGION

Let the classifier with $L$ number of layers be $F(x) = f_L(\sigma(f_{L-1}\sigma(\cdots\sigma(f_1(x))))) = f_{L:1}(x)$, where $x$ is the sample in the input space $\mathcal{X} \subset \mathbb{R}^{D_x}$ and $\sigma(\cdot)$ denotes the non-linear activation function[1]. For the $l$-th layer, $f_l(\cdot)$ denotes the linear operation and $f_{l:1}^i(\cdot)$ denotes the value of the $i$-th element of the feature vector $f_{l:1}(x) \in \mathbb{R}^{D_l}$. We define the DB for the $i$-th neuron of the $l$-th layer.

**Definition 1 (Decision Boundary (DB))** *The $i$-th decision boundary at the $l$-th layer is defined as,*

$$B_l^i = \{x | f_{l:1}^i(x) = 0, \quad \forall x \in \mathcal{X}\}.$$

---

[1]Although there are various activation functions, we only consider ReLU activation for this paper.

(a) PRS ratio        (b) Network A (Batch size: 2048) and network B (Batch size: 128)

Figure 2: (a) The number of PRS for the depth of each layer. (b) Training/Test accuracy and the PRS ratio on the penultimate layer of the networks with six convolution blocks (CNN-6) with batch size 2048/128. We select the networks at the 300th epoch and denote these two CNN-6 by Network A and B in the paper (PRS ratio of Network A: 0.99, and Network B: 0.007).

We note that the internal DB $B_l^i$ ($l < L$) divides the input space $\mathcal{X}$ based on the hidden representation of the $l$-th layer (i.e., existence of feature and the amount of feature activation). There are $D_l$ boundaries and the configuration of the DBs are determined by the training. As input samples in the same classification region are considered to belong to the same class, the input samples placed on the same side of the internal DB $B_l^i$ share the similar feature representation. In this sense, we define the internal DR, which is surrounded by internal DBs.

**Definition 2 (Decision Region (DR))** *Let $\mathbf{V}_l \in \{-1, +1\}^{D_l}$ be the indicator vector to choose positive or negative side of decision boundaries of the $l$-th layer. Then the decision region $DR_{\mathbf{V}_l}$, which shares the sign of feature representation, is defined as,*

$$DR_{\mathbf{V}_l} = \{x | sign(f_{l:1}(x)) = \mathbf{V}_l, \quad \forall x \in \mathcal{X}\}.$$

Figure 1 presents each training method (in Section 4) for VGG-16 with CIFAR-10 and the internal DBs/DRs of the penultimate layer ($f_{L-1:1}(x)$). We identify the proposed regularizer (A→ C) induces different configuration of DBs/DRs in the input space compared to the standard training (A→B).

## 2.2 Populated Region Set

It is shown that the number of DRs is related to the representation power of DNNs (Montúfar et al., 2014). In particular, the expressivity of DNNs with partial linear activation function is quantified by the maximal number of the linear regions and this number is related to the width and depth of the structure. We believe that although the maximal number can be one measure of expressivity, the trained DNNs with finite training data[2] does not handle the regions to solve the task. To only consider DRs that the network uses in the training, we devise the training-related regions where training samples are populated more frequently. We define the *Populated Region Set (PRS)*, which is a set of DRs containing at least one training sample. PRS can be utilized to analyze the relationship between the geometrical properties and the robustness of DNNs in a practical aspect.

**Definition 3 (Populated Region Set (PRS))** *From the set of every DRs of the model $f$ and given the dataset $\mathbf{X}$, the Populated Region Set for $l$-th layer is defined as,*

$$PRS(\mathbf{X}, f, l) = \{DR_{\mathbf{V}_l} | \mathbf{V}_l = sign(f_{l:1}(x)), \forall x \in \mathbf{X}\}.$$

*We can then define a Populated Region as a union of decision regions in PRS as,*

$$PR(\mathbf{X}, f, l) = \cup_{DR_{\mathbf{V}_l} \in PRS(\mathbf{X}, f, l)} DR_{\mathbf{V}_l}.$$

We note that the size of the PRS is bounded to the size of given dataset $\mathbf{X}$. When $|PRS(\mathbf{X}, f, l)| = |\mathbf{X}|$, each sample in training dataset is assigned to each distinct DR in the $l$-th layer. To compare the PRS among networks, we define the PRS ratio, $|PRS(f, \mathbf{X}, l)|/|\mathbf{X}|$, which measures the ratio between the size of the PRS and the given dataset. Figure 2 presents a comparison between two equivalent neural networks (A and B) with six convolution blocks (CNN-6) trained on CIFAR-10 varying only the batch size (2048/128). Figure 2 (a) presents the PRS ratio for the depth of layers in each model at the 300th epoch. We observe only the penultimate layer ($l = 8$) shows a different PRS ratio. Figure 2 (b) shows that the two networks have largely different PRS ratios with similar training/test accuracy. From the above observation and the fact that the penultimate layers are widely used as feature extraction, we only consider the PRS ratio on the penultimate layer in this paper.

---

[2]In general, the number of training data is smaller than the maximal number of the linear region.

# 3 ROBUSTNESS UNDER ADVERSARIAL ATTACKS

In this section, we perform experiments to analyze the relationship between the PRS ratio and the robustness. We evaluate the robustness of the network using the fast gradient sign method (FGSM) (Goodfellow et al., 2014), basic iterative method (BIM) (Kurakin et al., 2016) and projected gradient descent (PGD) (Madry et al., 2018) method widely used for the adversarial attacks. The untargeted adversarial attacks using training/test dataset are performed for the various perturbations ($\epsilon = 0.0313, 0.05, 0.1$).

## 3.1 EXPERIMENTAL SETUP

For the systematic experiments, we select three different structures of DNNs to analyze: (1) a convolutional neural network with six convolution blocks (CNN-6), (2) VGG-16 (Simonyan & Zisserman, 2014), and (3) ResNet-18 (He et al., 2016). We train basic models with fixed five random seeds and four batch sizes (64, 128, 512 and 2048) over three datasets: MNIST (LeCun & Cortes, 2010), F-MNIST (Xiao et al., 2017), and CIFAR-10 (Krizhevsky et al., 2009). For the extensive analysis on the correlation between the PRS ratio and properties of network, we extract candidates from each model with the grid of epochs. Then we apply the threshold for the test accuracy to guarantee the sufficient and similar performance. Finally, we obtain 947 models for analysis. The details for the network architecture and the selection procedure are described in Appendix A-C. We also perform an ablation study on the factors which affects the PRS ratio and the results are provided in Appendix E.

## 3.2 PRS AND ROBUSTNESS

First, we compare the two models (Network A and B in Figure 2) with similar test accuracy but different PRS ratio[3]. Figure 3 presents the results of robust accuracy under the FGSM, BIM (5-step) and PGD-20 (20-step) on $L_\infty$ with $\alpha = 2/255$. We identify Network B (low PRS ratio) is more robust than Network A (high PRS ratio) under all adversarial attacks.

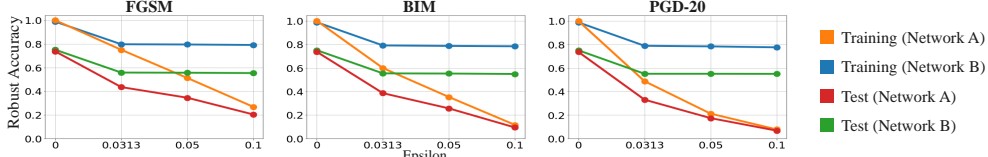

Figure 3: Robust accuracy under various adversarial attack methods on networks A and B. The x-axis indicates perturbation $\epsilon$ and the y-axis indicates the training/test robust accuracy.

From the above observation, we measure the PRS ratio and the robust accuracy in various models and datasets to verify the relationship between the PRS ratio and the robustness. Figure 4 presents the experimental results according to the various models under the PGD-20.

To quantify the relation, we calculate the coefficient of the regression line and perform significance test to validate the trend. We also provide the result of robust accuracy against the FGSM attack and AutoAttack (Croce & Hein, 2020) in Appendix F. In Figure 4, we identify the PRS ratio has an inversely correlated relationship with the robust accuracy in most cases. From this observations, we empirically confirm the PRS ratio is related to the robustness against adversarial attacks. To investigate the evidence that the low PRS ratio causes robustness for the gradient-based attack, we perform an additional analysis of failed attack samples. In the gradient-based attack, as the magnitude of the gradient is a crucial component to success, we observe the failure attack cases for Network A and B. We note the failed attack samples with non-zero gradients maintain the index of the largest logit as the true class after attack. To analyze the reason of failure, we examine the change of the logits under the adversarial attack in Figure 5 (a). To clarify the difference of the change of the logits between Network A and B, we select the examples of successful attack on Network A but failed attack on Network B. In Network B, the logits move on almost parallel direction, which causes the predicted label to be maintained as the true class.

---

[3]We note different PRS ratios are obtained by different batch size of Network A (2048) and B (128).

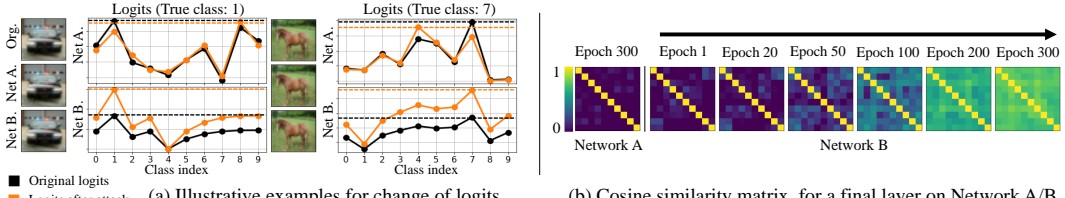

Figure 4: Relationship between the PRS ratio and robust accuracy attacked by PGD ($\epsilon$ : MNIST = 0.3, F-MNIST = 0.1, and CIFAR10 = 0.0313 on $L_\infty$ norm). The colored dots are for the independent models described in Appendix A. The colored dashed lines indicate the trend for each dataset.

Figure 5: (a) Illustrative examples of attacked samples on Network A and B which is failed on B, and the corresponding logits before/after the attack. (PGD-20 on $L_\infty$ with $\epsilon = 0.0313$). After the attack, the logits move on almost parallel direction with the original logits in Network B. More examples are provided in Appendix C. (b) Cosine similarity (CS) matrix for a final layer on Network A/B. As the epoch increases, the CS for each parameter increases on Network B.

|  | CNN-6 | | VGG-16 | | ResNet-18 | |
|---|---|---|---|---|---|---|
| Dataset | Coef | P-val | Coef | P-val | Coef | P-val |
| MNIST | -0.76 | 5.17E-29 | -14.7 | 2.70E-08 | -2.88 | 1.36E-11 |
| F-MNIST | -0.58 | 2.43E-46 | -3.38 | 4.11E-19 | -2.24 | 1.66E-16 |
| CIFAR-10 | -0.65 | 4.35E-54 | 0.28 | 2.92E-01 | -1.29 | 4.20E-14 |

Table 1: Coefficient (Coef) and P-value (P-val) of the regression analysis between the PRS ratio and cosine similarity.

To explain the parallel change of the logit vector, we hypothesize the DBs corresponding to each class node have similar configuration in the input space. However, it is intractable to measure the similarity between DBs in the network due to the highly non-linear structure and the high dimensional input space. To simplify our hypothesis, we only measure the cosine similarity between the parameters which map the features on the penultimate layer to logits (i.e., final layer). Figure 5 (b) presents the similarity matrices for Networks A and B. When we compare the matrices between the two models at the 300th epoch, we identify Network B (low PRS ratio) has higher cosine similarity between each parameter in the final layer. We note that the cosine similarity between each parameter in the final layer can be considered as the degree of parallelism for the normal vectors in the linear classifier. We also confirm the decrease of the PRS ratio is aligned with the increase of the similarity of parameters in Figure 5 (b) on Network B, when we consider the graph in Figure 2. To verify the relationship between PRS ratio and the cosine similarity, we measure the PRS ratio and the cosine similarity between each parameter for all 947 candidate models. Table 1 shows the results of the regression analysis for the relationship between the PRS ratio and cosine similarity. We identify the PRS ratio has an inverse correlation for the cosine similarity between each parameter in the final layer.

## 3.3 PRS AND TEST SAMPLES

When we regard the model as a mapping function from the input space to the feature space, if the majority of samples from the test dataset are assigned to the training PR, the model can be considered

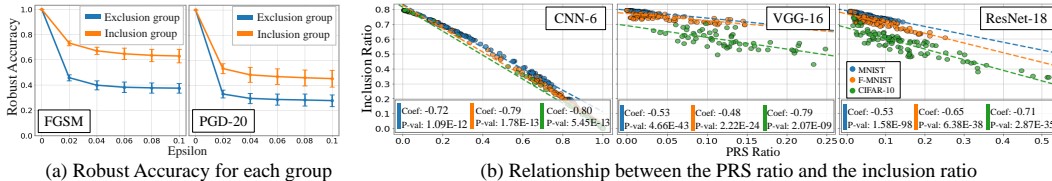

(a) Robust Accuracy for each group      (b) Relationship between the PRS ratio and the inclusion ratio

Figure 6: (a) Test accuracy under adversarial attacks for inclusion/exclusion groups for CNN-6 on CIFAR-10. The blue/orange line indicates the exclusion/inclusion group, respectively. (b) Relationship between the PRS ratio and the inclusion ratio for various models and datasets.

to be learned the informative and general concept of feature mapping. However, it is non-trivial to guess which decision will appear when the test sample is mapped to out of the training PR. To investigate the differences between the test samples which are included and excluded in the training PR, we evaluate the test accuracy under adversarial attack for each group. For a comparison, we divide both the inclusion and exclusion groups with 1k correctly predicted test samples. Figure 6 (a) shows the robust accuracy under the FGSM and PGD-20 on $L_\infty$. Although the robust accuracy of each test group decreases as the epsilon becomes larger, we observe the inclusion group is more robust against both attacks compared to the exclusion group. More experimental results are provided in Appendix H. Figure 6 (b) presents the results of the correlation for the relationship between the PRS ratio and the inclusion ratio of the test samples for the training PR. We compute the inclusion ratio as the ratio of the test samples mapped to the training PR, and identify the PRS ratio and the inclusion ratio have inversely correlated relationship. As we previously verify that the included test samples show high robustness, we empirically confirm that the low PRS ratio is related to the robustness under adversarial attacks.

## 3.4 PRS AND TRAINING SAMPLES

From Section 3.3, we empirically observe that the vulnerability of individual test samples is related to PRS defined by training samples. In this section, we categorize the element of the PRS to expand this relationship. At first, we define the major DR for each class $c$ which includes the majority of training samples.

**Definition 4 (Major Region (MR))** *Let the training dataset with class $c$ $\mathbf{X}_c$ and the classifier $f$. The major region for $l$-th layer and class $c$ is defined as,*

$$MR_{l,c} = \operatorname*{argmax}_{DR_{V_l} \in PRS(\mathbf{X}_c, f, l)} |\{x \,|\, sign(f_{l:1}(x)) = \boldsymbol{V}_l, \forall \, x \in \mathbf{X}_c\}|.$$

We note that $MR_{l,c}$ denotes the DR and $MR_{l,c} \cap \mathbf{X}_c$ represents the training samples which populate in $MR_{l,c}$. We refer the remained DRs (i.e., not MR) as the extra regions (ER).

### 3.4.1 COMPARISON OF MR AND ER

At first, we observe the distribution of training samples for the type of region corresponding each class in VGG-16 trained with CIFAR-10. Figure 7 (a) depicts the distribution of training samples for MR (sky blue bar and black dashed line) and ER for each class. Although the training samples are distributed the various regions, we identify that MR exists for each class. To compare the characteristics of samples populated each region, we randomly selected 10k training samples from MR and ER. We perform adversarial attack for selected samples and measure the confidence of the prediction (logit value for the target class). Figure 7 (b) and (c) show the robust accuracy and confidence for MR and ER, respectively. We empirically verify the training samples in MR have higher adversarial robustness and the network predicts these samples with high confidence.

From the empirical observations that samples belonging to MR are relatively robust, we hypothesize that samples located closer to center of MR tends to be more robust. To represent the center of MR, we assume the distribution of feature vector $f_{l:1}(x)$ for populated training samples $\forall x \in \mathbf{X}_c$ is gaussian distribution.

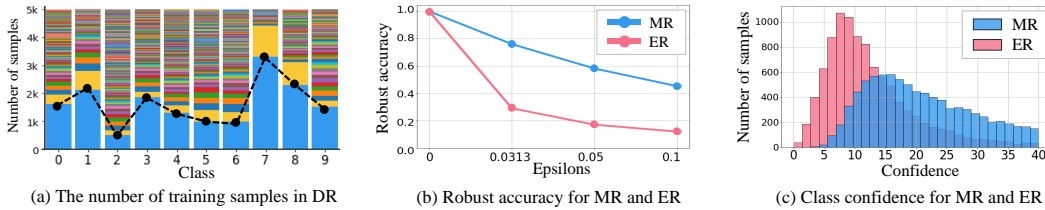

Figure 7: Experimental results under VGG-16 on CIFAR-10. (a) The number of training samples populated in DRs for each class. The maximum number of samples for each class is 5k and the sky blue bar indicates each MR. (b) Robust accuracy under PGD-20 attacks on $L_\infty$ for the samples in MR and ER. (c) Class confidence for the samples in MR and ER.

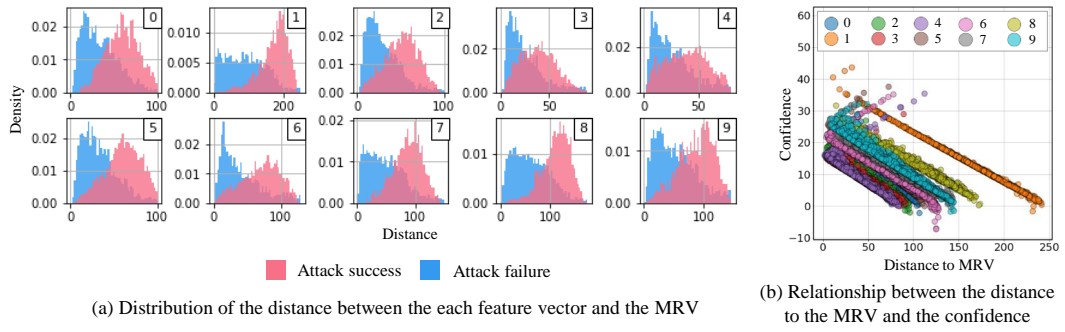

(a) Distribution of the distance between the each feature vector and the MRV

(b) Relationship between the distance to the MRV and the confidence

Figure 8: Experimental results under VGG-16 on CIFAR-10. (a) Distribution of distance to MRV for training samples. The blue/red bar indicates the failed/success attack samples and the white box in upper right presents each class. (b) Relationship between the distance to MRV and the confidence. The colored dots represent samples which are vulnerable under adversarial attack per each class.

**Definition 5 (Major Region Mean Vectors (MRV))** *Let the major region for class $c$ of $l$-th layer $MR_{l,c}$ and the classifier $f$. The mean vector of major region is defined as,*

$$MRV_{l,c} = \frac{1}{|MR_{l,c} \cap \mathbf{X}_c|} \sum_{x \in MR_{l,c} \cap \mathbf{X}_c} f_{l:1}(x).$$

To verify our hypothesis, we measure Euclidean distance between MRV and training samples with success/failure of adversarial attack. From Figure 8 (a), we identify the samples far from MRV tend to be vulnerable for the adversarial attack in the class. Furthermore, in Figure 8 (b), we identify the inversely correlated relationship between the confidence and the distance to MRV for the failed attack samples.

## 4 ROBUST LEARNING VIA PRS REGULARIZER

In previous Sections, we empirically verify that PRS is related to the adversarial robustness. In particular, (1) inclusion of MR, and (2) distance to MRV are highly related to vulnerability of individual samples. From these insights, we devise a novel regularizer leveraging the properties of PRS to improve the adversarial robustness.

### 4.1 REGULARIZER VIA PRS

At first, we design the regularizer to reduce the distance between feature vector and MRV. To guarantee the quality of feature representation which constructs plausible PRS, we utilize the warm-up stage for the classifier. In the warm-up stage, the classifier is trained with cross-entropy loss function $\mathcal{L}_{ce}$ during the $t$-th epoch. After warm-up stage, we construct the MRV for each class and use it after $t$-th epoch. Let $\{(x_i, y_i)\}_{i=1}^N$ be a training dataset and $f$ be a classifier. The regularizer for MRV is

| Model | Method | Robust Acc | Test Acc | PRS Ratio | Time/Epoch (s) |
|---|---|---|---|---|---|
| CNN-6 | Standard | 38.82±2.73 | 77.92±0.34 | 0.101±0.010 | 8.92±0.32 |
| | AT | 53.79±0.42 | 70.65±0.17 | 0.099±0.001 | 38.63±0.09 |
| | $\mathcal{L}_{MR}$ (Ours) | 53.59±0.07 | 80.30±0.72 | 0.018±0.003 | 9.46±0.13 |
| | $\mathcal{L}_{PRS}$ (Ours) | **54.52±0.61** | **80.35±0.11** | 0.017±0.002 | 9.78±0.07 |
| VGG-16 | Standard | 39.94±1.28 | **80.28±0.24** | 0.115±0.012 | 11.63±0.18 |
| | AT | 58.18±0.13 | 75.22±0.05 | 0.069±0.002 | 81.48±0.23 |
| | $\mathcal{L}_{MR}$ (Ours) | 60.42±0.36 | 78.61±0.15 | 0.038±0.007 | 12.15±0.21 |
| | $\mathcal{L}_{PRS}$ (Ours) | **61.08±0.88** | 79.09±0.35 | 0.031±0.005 | 12.63±0.11 |
| Resnet-18 | Standard | 33.48±0.08 | **76.96±0.15** | 0.065±0.001 | 12.11±0.32 |
| | AT | **50.65±0.20** | 73.03±0.05 | 0.046±0.004 | 58.96±0.15 |
| | $\mathcal{L}_{MR}$ (Ours) | 49.31±0.65 | 76.51±0.08 | 0.061±0.003 | 12.97±0.29 |
| | $\mathcal{L}_{PRS}$ (Ours) | 50.54±0.10 | 76.44±0.18 | 0.038±0.004 | 13.02±0.35 |

Table 2: Comparison of robust and test accuracy under PGD-20 attacks on $L_\infty$ for CIFAR-10.

defined as,

$$\mathcal{L}_{MRV} = \frac{1}{N} \sum_{i=1}^{N} \left( MRV_{l,y_i} - f_{l:1}(x_i) \right)^2. \tag{1}$$

We note that because $\mathcal{L}_{MRV}$ reduce the distance to MRV in the feature space, the arbitrary sample can have the opportunity for inclusion of MR. However, it is non-trivial to guarantee for inclusion based on Euclidean distance, because the feature vector is encoded (1) in the high dimensional space, and (2) with highly non-linear representations. To ensure that the training samples are included into MR, we devise additional regularizer through Hamming distance.

$$\mathcal{L}_{ham} = \frac{1}{ND_l} \sum_{i=1}^{N} \sum_{j=1}^{D_l} \text{sign}\left( (MRV_{l,y_i})_j \right) \oplus \text{sign}\left( f_{l:1}(x_i)_j \right) \tag{2}$$

where $D_l$ is dimension of $l$-th layer, and $\oplus$ is the exclusive operator which returns zero when the target and prediction are identical and one otherwise. Finally, we define PRS regularizer $\mathcal{L}_{PRS}$ with the weighted sum of $\mathcal{L}_{ce}$, $\mathcal{L}_{MRV}$ and $\mathcal{L}_{ham}$.

$$\mathcal{L}_{PRS} = \lambda_1 \cdot \mathcal{L}_{ce} + \lambda_2 \cdot \mathcal{L}_{MRV} + \lambda_3 \cdot \mathcal{L}_{ham} \tag{3}$$

where $\lambda_1$, $\lambda_2$, and $\lambda_3$ are hyperparameters. We perform simple grid search to set hyperparameters and use $\lambda_1 = 0.2$, $\lambda_2 = 0.8$, and $\lambda_3 = 1$ for remained experiments. We denote the loss function with $\lambda_3 = 0$ by $\mathcal{L}_{MR}$.

## 4.2 EXPERIMENTAL RESULTS

To verify the effectiveness of our method, we apply the regularizer to various models for the classification task on CIFAR-10. We set standard training ($\mathcal{L}_{ce}$) and adversarial training (AT) based on PGD-20 on $L_\infty$ as the baselines. We freeze the parameters of the final layer for the trained model with standard training in warm-up stage ($t = 50$), and observe the effect of the change of PRS for robust and test accuracy. Figure 1 depicts the training procedure for each training method and trained DBs and DRs which represent the state of PRS. In Table 2, we verify that the proposed method can improve the robust accuracy while maintaining the test accuracy. We note that the proposed PRS regularizer does not use the adversarial examples to improve the adversarial robustness. It means our method can have strength in the perspective of computation time. We further investigate the change of PRS after training for each method. In Figure 9, we identify the both proposed regularizers can increase the number of populated training samples in MR compared to the standard training (i.e., reduce of PRS ratio). Interestingly, we confirm the similar result for change of PRS in AT. Although AT and PRS regularizer are not related directly, we consider this phenomenon can be empirical evidence to support relationship between PRS and adversarial robustness. We also provide the results for different networks and large dataset in Appendix F.2.

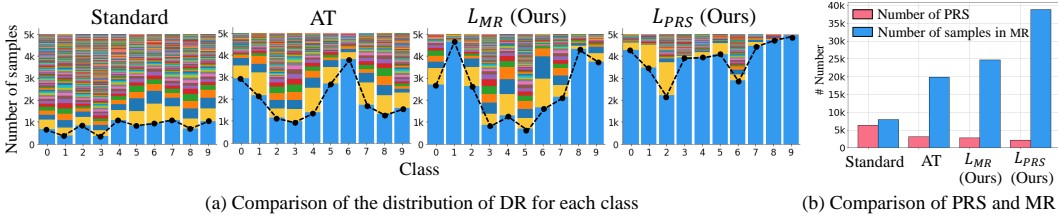

(a) Comparison of the distribution of DR for each class

(b) Comparison of PRS and MR

Figure 9: Experimental results under VGG-16 on CIFAR-10. (a) The distribution of DR for each class in various training methods. The sky blue bar per class indicates each MR. (b) The number of PRS and samples in MR.

## 5 RELATED WORK

The adversarial attack which reveals the vulnerability of DNNs, is mainly used to validate the reliability of the trained network. As an early stage for adversarial attacks, the fast gradient sign method (FGSM) (Goodfellow et al., 2014) based on the gradient with respect to the loss function and the multi-step iterative method (Kurakin et al., 2016) are proposed to create adversarial examples to change the model prediction with a small perturbation. Recently, many studies on effective attacks in various settings have been performed to understand the undesirable decision of the networks (Chen et al., 2020; Madry et al., 2018; Shaham et al., 2018). In terms of factors affecting robustness, Yao et al. (2018) provides evidence to argue that training with a large batch size can degrade the robustness of the model against the adversarial attack from the perspective of the Hessian spectrum. With increasing interest in the expressive power of DNNs, there have been several attempts to analyze DNNs from a geometric perspective (Choromanska et al., 2015; Dauphin et al., 2014). In these studies, the characteristics of the decision boundary or regions formulated by the DNNs are mainly discussed. Montúfar et al. (2014) show the cascade of the linear layer and the nonlinear activation organizes the numerous piece-wise linear regions. They show the complexity of the decision boundary is related to the maximal number of these linear regions, which is determined by the depth and the width of the model. Xiong et al. (2020) extend the notion of the linear region to the convolutional layers and show the better geometric efficiency of the convolutional layers. Compared the previous work, our work focuses on the practical decision region which the trained network actually utilizes, interpreting the vulnerability of DNNs with the geometry is another important topic. Croce et al. (2019) discusses the relationship between the max margin of linear regions (LRs) and the robustness in ReLU networks. In particular, they utilize the perpendicular distance to the linearized internal/final decision boundaries (DBs) which comprise the LRs in the input space. In the bounded input space, increasing the distance to DBs expands the volume of LRs, and it eventually reduces the number of LRs. But, as our work mainly focuses on the number of PR on the penultimate layer rather than LRs, $L_{PRS}$ is not limited to the partial linear relation between the input samples and internal DBs, and it can reduce the PRS ratio directly.

## 6 CONCLUSION

In this work, we propose the novel concept *Populated Region Set (PRS)* to derive the relationship between the geometrical properties of DNNs and robustness in a practical setting. From systematic experiments with the proposed concept, we observe the empirical evidences that the PRS is related to the adversarial robustness of DNNs: (1) The network with the a low PRS ratio shows high robustness against the gradient-based attack compared to the network with a high PRS ratio. In particular, the model with a low PRS ratio has a higher degree of parallelism for the parameters in the final layer, which can support robustness. (2) The network with a low PRS ratio includes more test samples in the training PR. We empirically verify this inclusion ratio is related to robustness from the observation that included test samples are more robust than excluded test samples. (3) The proposed PRS regularizer can improve the robustness without adversarial examples. We verify the proposed method is valid for various network architecture. We also observe the adversarial training reduces the PRS ratio with the improvement of robust accuracy. It can be considered as another empirical evidence to support the relationship between PRS and adversarial robustness. Our work provides the insight of the geometrical interpretation of robustness in perspective of decision region. We expect the concept of PRS would contribute to the improvement of robustness in DNNs.

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
