# OpenReview forum: "On the Relationship Between Adversarial Robustness and Decision Region in Deep Neural Networks"
_ICLR.cc/2023/Conference — Submitted to ICLR 2023_

### Official Review · Reviewer_o1vx · 2022-10-23

**Confidence:** 4
**Correctness:** 3
**Technical Novelty And Significance:** 3
**Empirical Novelty And Significance:** 3
**Recommendation:** 6

**Clarity, Quality, Novelty And Reproducibility:**

The paper provides novel method and is written in good quality. Some technical details are not clearly explained.

**Strength And Weaknesses:**

## Strength
1. This paper provides novel concept on PRS and PRS ratio, and make connection between the proposed concept and adversarial robustness
2. Thorough experiments are conducted to emperically support the claim on the relationship between PRS and robustness
3. The proposed regualrization indicates the potential benefit PRS can bring to understanding and preventing adversarial attack

## Weakness
Some technical details of the proposed method are not clearly explained. For example:
1. How exactly is PRS computed given a DNN model? Is the computation method scalable to a larger/deeper model or a larger dataset?
2. How is the gradient of the Hamming distance regularization is Eq.(2) computed in the optimization?
3. As the model is being updated with the proposed regularization, will MRV change in the process? Does MRV need to be recomputed during the training process, and if so what would be the cost of the computation?

**Summary Of The Paper:**

This paper prooses the novel concept of populated region set, and make connection between the proposed PRS ratio and the adversarial robustness of the model. The paper further propose PRS regualrization that can improve adversarial robustness without adversarial training.

**Summary Of The Review:**

Generally the paper provides novel and interesting insight on understanding the adversarial robustness of DNN models. However the lack of technical details make it hard to assess the correctness and practicalness of the proposed method, Thus I would recommend a weak rejection for now.

## Post rebuttal
I believe the response from the author clearifies my doubts. Given the author response and revision I'm increasing my score.

---

> ### Author Response · Authors · 2022-11-11
> **Response to Reviewer o1vx**
>
> We deeply appreciate your efforts in reviewing our paper, as well as the constructive comments. We have revised the manuscript by faithfully reflecting your comments. We respond to your comments in the following.
>
> ---
>
> **Q1. How exactly is PRS computed given a DNN model? Is the computation method scalable to a larger/deeper model or a larger dataset?**
>
> ---
> - To compute $PRS(\mathbf{X}, f, l)$ in the Definition 3, we need to stack the sign of feature vector for training dataset $\mathbf{X}$ at the target layer $l$.
>  For each data $x\in\mathbf{X}$, we feed $x$ to the model $f$ to obtain the sign of feature vector at the target layer $l$. Then we take the unique over the stacked sign vectors to acquire the populated region set, $PRS(\mathbf{X},f,l)$.
>
> - As a result, we need 2 steps for the computation : (1) collecting sign vectors on the specific layer for entire training smamples, and (2) applying unique function for the collected sign vectors.
>
> - We think that the computation of PRS can be scalable to larger/deeper model, because the PRS is defined by collecting the decision regions where the given dataset are actually populated. As a result, the computaion of PRS is linearly propotional to the size of the model.
>
> - However, the current version of computation seems difficult to be scalable for larger dataset. Because the unique function for computing PRS requires the expensive cost for larger dataset. To alleviate the computation bottleneck, we can consider a sampling strategy for each class by constructing a subset of the training dataset, which can be an alternative to approximate PRS.
>
> ---
>
> **Q2. How is the gradient of the Hamming distance regularization is Eq.(2) computed in the optimization?**
>
> ---
>
> - As the sign function to compute the activation pattern is not differentiable, we utilize the Hyperbolic tangent function (tanh) to approximate the sign function, which can have a derivative. Then we compute the Euclidean distance between approximated sign vectors. Finally, the Hamming loss (implementation version) is defined as,
>
> $$\mathcal{L_ham} = \dfrac{1}{ND_{l}} \sum_{i=1}^{N} \sqrt{\sum_{j=1}^{D_{l}} \big(k\sigma \big((MRV_{l, y_{i}})_j \big) - k\sigma \big( f_\{l:1}(x_i)_j \big)\big)^2} $$
>
> where $\sigma$ is tanh function and $k$ is the constant to control the slope of tanh function for the approximation of the sign function. In the experiment, we used $k=5$.
>
> ---
>
> **Q3. As the model is being updated with the proposed regularization, will MRV change in the process? Does MRV need to be recomputed during the training process, and if so what would be the cost of the computation?**
>
> ---
>
> - In the experiments of the main paper, we adopt the warm-up strategy which computes the $MRV$ once after $t$-th epoch ($t=50$) as we empirically observe that the $MRV$ is not significantly changed after warm-up stage.
>
> - However, if $MRV$ needs to be recomputed every epoch, it requires "Epoch$\times N$" time where $N$ is the computation time for MRV. For example, on VGG-16 under CIFAR-10, $N=10.429$ seconds on
> a single Quadro RTX 6000 gpu.
>
> ---
> We sincerely thank you for the insightful comments. Please let us know if there is anything else we should address, or additional questions.

---

> > ### Comment · Reviewer_o1vx · 2022-11-16
> > **Thanks for the clearification**
> >
> > I would like to thank the author for the clearification made in the response and the revision. I think the notation and the methodology are clear.
> >
> > I have also looked into the discussion between Reviewer wNGM and the author. I think the author makes a good response on the concern of AA robustness. Though PRS regularization does not achieve super-high robustness on AA, the robustness improvement on both clean training and adversirial training shows the potential of improving robustness by inducing low PRS ratio. Together with the other observations provided by the author, I think there is a clear relationship between the PRS ratio and the robustness of the model, which is a good contribution to the field. To this end I'm leaning to accept this paper.

---

> > > ### Author Response · Authors · 2022-11-17
> > > **Thank you for the response**
> > >
> > > We are really happy to hear that our response resolved your questions.
> > > We are grateful for your insightful comments.
> > >
> > > If you have any further questions or suggestions, please do not hesitate to let us know.
> > >
> > > Thank you very much,
> > >
> > > Authors

---

### Official Review · Reviewer_wNGM · 2022-10-23

**Confidence:** 4
**Correctness:** 2
**Technical Novelty And Significance:** 2
**Empirical Novelty And Significance:** 3
**Recommendation:** 3

**Clarity, Quality, Novelty And Reproducibility:**

## Clarity
* The illustration and structure of the paper aid the clarity of the paper.
* Section 1 contains terms that are not yet defined like PRS ratio, PRS inclusion ratio of test examples etc.
* However, the writing could be improved. It is not a major drawback but probably something to look at for future versions.

## Quality and Novelty
*  The authors have made a clear hypothesis and tried to validate with experiments. I find the way the flowchart of the paper quite good. The problem is whether the experiments bear the results and it does not.

## Reproducibility
* I do not have any major concerns.

**Strength And Weaknesses:**

## Strength

* The concept of PRS is interesting and well motivated.
* The design of the regulariser is also well motivated.
* The figures and illustrations are easy to understand and the mathematical formalisations are minimal and kept to the point. Overall, the authors have tried to present a coherent story.

## Weaknesses
* The message of the paper is not borne by the experimental results. In particular, AA robust accuracy is always zero.
* It seems that certain networks have not been evaluated under AA and all AA results are relegated to the appendix.

**Summary Of The Paper:**

The paper looks at he notion of _Populated Region Set (PRS)_ which are decision regions with at least one training example in them. The central message of the paper is that networks with lower number of PRS have better robustness.

**Summary Of The Review:**

The main drawbacks of the paper  are the following

* The first red flag should have been that for $\epsilon=0.1$, a CNN network trained via standard training has 80% robust accuracy in Figure 3. This is the main result the authors use to justify that low PRS ratio increases robustness. However, a quick look at Appendix F shows that both Model A and Model B have less than 1% robust accuracy. This shows that it is not that low PRS ratio models are more robust just that, it is harder to find adversarial examples possibly due to ill-conditioning.

*Again Figure 6 shows a similar kind of behaviour as Figure and does not include measurements with AutoAttack.

*Finally table 2 also doesn't contain AA. In fact, even the PGD attack is fairly weak containing only 20 steps. This can also call into question the PRS regularisers are indeed more robust than ST and whether PRS regulariser+AT is more robust than AT.

I did find Appendix G quite interesting, as it does suggest a connection between PRS ratio and robustness but this is the only experiment that is not confounded by the possibility of weak attack.

Given the above, I am inclined to support rejection unless the authors convince me otherwise.

---

> ### Author Response · Authors · 2022-11-11
> **Response to Reviewer wNGM (2/2)**
>
> ---
>
> **Q4. Again Figure 6 shows a similar kind of behaviour as Figure and does not include measurements with AutoAttack.**
>
> ---
>
> - From the answer in Q1, as the robust accuracy under AA on "Standardly trained model" is almost zero, it is not comparable the robust accuracy between the inclusion and the exclusion group (Figure 6 (a)).
>
> - Figure 6 (b) shows the relationship between the PRS ratio and the inclusion ratio. So it is not related to robust accuracy under AA.
>
> ---
>
> **Q5. Finally table 2 also doesn't contain AA. In fact, even the PGD attack is fairly weak containing only 20 steps. This can also call into question the PRS regularisers are indeed more robust than ST and whether PRS regulariser+AT is more robust than AT.**
>
> ---
>
> - The experimental results of the PRS regularizer under AA are provided in Appendix F.2 (see Table 6). In this experiment, although the proposed regularizers ($L_{PRS}$) cannot beat the adversarial training (AT), our proposed method shows the significant improvement of the robustness compared to the standard training. We also confirm that $L_{PRS}$+$AT$ still shows better results than AT to alleiviate the drop of clean accuracy.
>
> - We also note that our PRS regularizer can improve the robust accuracy without adversarial examples which require the expensive computation cost. Please see 'Time/Epoch (s)' column in Table 2 of the main paper and Table 6 in Appendix F.2. This can be an additional strength against AT.
>
> - Although the PRS regularizer is proposed as an application which utilizes our empirical study between PRS and robustness, we think that the concept of PRS cannot be limited to just the regularizer. We believe that PRS can be a starting point to provide novel insight for the robustness of the deep neural networks.
>
> ---
>
> **Q6. I did find Appendix G quite interesting, as it does suggest a connection between PRS ratio and robustness but this is the only experiment that is not confounded by the possibility of weak attack.**
>
> ---
>
> - We can also identify the mentioned relationship in not only Appendix G but also Figure 9 of the main paper. We confirm that applying AT to the network can reduce the PRS ratio (Figure 9 (b)) and increase the number of samples in MR (Figure 9 (b)). We believe that these observations can provide additional evidences to support the relationship between PRS and robustness.
>
> ---
> We sincerely thank you for the insightful comments. Please let us know if there is anything else we should address, or additional questions.

---

> > ### Comment · Reviewer_wNGM · 2022-11-11
> > **Robustness is a property of the ML model against a threat model**
> >
> > I thank the authors for their response.
> >
> > I would like to clarify my explanation for why robustness against AA is necessary to talk about robustness of the network. The main thesis of the paper is that certain structures in the decision regions wrt to the training data can help us infer about the *robustness of the model*. Here, the concept of *robustness of the model* is defined in terms of the closeness of the decision boundary to the data (more precisely what fraction of the data lies close to the decision boundary). If the authors agree on this definition of robustness, then there is no difference between AA and PGD-20 except that they are both different approximations to the *true robustness* and hence the higher robust error of the two approximations are close to the true quantity. Then, I believe there is no significance in claiming robustness against PGD-20 but failing utterly against AA. All it says it that 20 PGD steps within the epsilon ball are not sufficient to find the decision boundary even though the decision boundary intersects with the epsilon ball (as shown by AA).
> >
> > If I am misunderstanding something here, I would appreciate a clarification.

---

> > > ### Author Response · Authors · 2022-11-15
> > > **Response to Reviewer wNGM**
> > >
> > > - We sincerely thank you for the thoughtful response. We also agree that the robustness of the deep neural networks is described in a perspective of the closeness between the decision boundary (DB) on final layer and input samples.
> > >
> > > - We think that the models with standard training seem to be more vulnerable under certain attacks (e.g., AA), because they are not trained by directly increasing the margin to the final DB. Among the models with standard training, the robust accuracy under AA was closed to 0, but the model with our proposed PRS regularizer shows the significant improvement in robust accuracy even under AA (0.21% -> 12.4%, Table 6 in the Appendix F.2).
> > >
> > > - It denotes that the properties of feature embeddings described by PRS (we note that the PRS is computed in the penultimate layer) can be an alternative to understand the robustness of the network. In Table 6, we can also confirm that the $L_{PRS}+AT$ acheives higher performance by mitigating the degradation of clean accuracy than $AT$.
> > > In other words, using $L_{PRS}+AT$, which directly reduces PR, can be more powerful than using only $AT$. That is, we think that a method which directly reduces PR can help to improve robustness.
> > >
> > > - We also provide a toy example to clarify our concepts for the internal DBs and DRs under adversarial attack in Appendix J. We can identify that the configuration of PRS (on the penultimate layer) affects to the margin of final DB, and it can cause improvement of the robustness. We note that in this example, we did not utilize the advesrial examples for training. We believe that our observations can be an evidence which the concept of PRS can be a novel tool to understand the relationship between feature embeddings and robustness of the network.

---

> > > > ### Comment · Reviewer_wNGM · 2022-11-23
> > > > **Model's robustness or attack's efficiency**
> > > >
> > > > I am still convinced that it is not sound to use PGD to talk about the robustness of the model **when we know that the model is not at all robust** (courtesy of AA). All the high or low PGD accuracy says is whether PGD, as an attack, succeeds or not and it does not say whether the model is robust.

---

> ### Author Response · Authors · 2022-11-11
> **Response to Reviewer wNGM (1/2)**
>
> We sincerely appreciate your efforts in reviewing our paper, as well as the constructive comments. We have revised the manuscript by faithfully reflecting your comments. We respond to your comments in the following.
>
> ---
>
> **Q1. The message of the paper is not borne by the experimental results. In particular, AA robust accuracy is always zero.
> It seems that certain networks have not been evaluated under AA and all AA results are relegated to the appendix.**
>
> ---
>
> - We are sorry for the absence of description for AutoAttack (AA) in detail. As can be seen from the AA results for "Standardly trained model" in Robust bench [1], the models show zero robust accuracy against AA known as the most powerful adversarial attack (see CIFAR-10 with $l_\infty$ or $l_2$). In our experiment, as entire candidates in Table 2 of Appendix A are standardly trained, the robust accuracy under AA is mostly close to zero. As a result, it is not comparable the trend of relationship between PRS ratio and robustness under AA. As an alternative, we provide the experiment for various defense methods under AA in Appendix F.2 which you mentioned. We have revised this contents to reveal in the main paper.
>
> [1] https://robustbench.github.io/
>
>
> ---
>
> **Q2. Section 1 contains terms that are not yet defined like PRS ratio, PRS inclusion ratio of test examples etc.  However, the writing could be improved. It is not a major drawback but probably something to look at for future versions.**
>
> ---
>
> - We are sorry for mentioning the notions such as PRS ratio before discribing the definition which can cause the misleading. We have revised the notions in Section 1 to clarify the contents without the definitions. Thank you for the constructive comments to improve the quality of the manuscript.
>
> ---
>
> **Q3. The first red flag should have been that for \epsilon=0.1 , a CNN network trained via standard training has 80% robust accuracy in Figure 3. This is the main result the authors use to justify that low PRS ratio increases robustness. However, a quick look at Appendix F shows that both Model A and Model B have less than 1% robust accuracy. This shows that it is not that low PRS ratio models are more robust just that, it is harder to find adversarial examples possibly due to ill-conditioning.**
>
> ---
>
> - The robust accruracy mentioned in Figure 3 ("80%") is the result under PGD-20 on training samples. However, the robust accuracy less than 1% in Appendix F is the result under AA on test samples. So the type of attack and the type of samples are different.
>
> - Except for AA, in various conditions (e.g., various structures), we identify the significant relationship between PRS and robustness. The experiments of PRS regularizer can also support that the PRS is clearly related to the robustness regardless of the type of adversarial attacks (i.e, ill-conditioning of advesarial examples). As a result, we confirm that the more robust models tend to have the lower PRS ratio.

---

### Official Review · Reviewer_mBon · 2022-10-24

**Confidence:** 4
**Correctness:** 3
**Technical Novelty And Significance:** 3
**Empirical Novelty And Significance:** 3
**Recommendation:** 5

**Clarity, Quality, Novelty And Reproducibility:**

Clarity: This overall paper is clearly written and well organized. I find it easy to follow. However, some definitions are confusing.
Quality: This paper is technically sound.
Novelty: The novelty of this paper is high.

**Strength And Weaknesses:**

Strength: 1. The concept of the Populated Region Set (PRS) is novel and interesting.
2. The experiments are convincing.

Weaknesses: 1. Lack of theoretical analysis to verify the claims.
2. Some definitions are confusing. For example, in Definition 2 (Decision Region (DR)), if the i-th layer has \textbf{only one neuron}, what are $D_\ell$ and $|DR_{V_\ell}|$? From the definition, it seems that $D_\ell =1$ and $|DR_{V_\ell}|=2$ in this case, but this seems strange. Please give one small example for each definition to make things clear.

**Summary Of The Paper:**

In this paper, the authors introduced the concept of Populated Region Set (PRS) to characterize the complexity of DNNs and build the correction between low PRS ratio and high robustness of models via several experiments.

**Summary Of The Review:**

In summary, the novelty of this paper is high, and the concept of the Populated Region Set (PRS) is novel and interesting. However, my main concern is that doesn't provide theoretical analysis, which makes the contribution limited. Also, some definitions are confusing at this point.

---

> ### Author Response · Authors · 2022-11-11
> **Response to Reviewer mBon**
>
> We deeply appreciate your efforts in reviewing our paper, as well as the constructive comments. We have revised the manuscript by faithfully reflecting your comments. We respond to your comments in the following.
>
> ---
>
> **Q1. Lack of theoretical analysis to verify the claims.**
>
> ---
>
> - In general, the adversarial robustness is interpreted by the concept of margin in the deep neural networks [1,2,3]. We conjecture that our PRS ratio is related to the concept of margin.
>
> - In the bounded input space, increasing the distance to internal DBs expands the volume of internal DRs, and it eventually reduces the number of DRs. To verify our hypothesis, we have provided a 2D binary classification toy example with a simple fully-connected ReLU network (2-200-200-2) with standard training and $L_{PRS}$ in Appendix J. In Figure 23, we provide the visualization of (1) the final decision boundary, (2) result of adversarial attack (FGSM), (3) the linear regions [4], and (4) PRS between the model with standard training and $L_{PRS}$.
>
> - At first, in the first and second columns, we identify that the models have different configuration of the final DB and different robust accuracy, but with similar train accuracy. However, in the third and fourth column, we can observe the siginficant difference between these two models. The PRS regularizer tends to merge the linear regions and the model with $L_{PRS}$ have finally the small number of PRS in the input space compared to the model with standard training. And the model with $L_{PRS}$ seems to have the larger margin between training samples and the final DB, compared to the standardly trained model.
>
> - From this observation, we identify that $L_{PRS}$ merges PRs directly, and this aligns with the behavior leading to the increase of margin to internal DBs. We believe that this empirical observation can be a bridge to connect the concept of margin and PRS. It also can be plausible starting point for the theoretical analysis for the relationship between PRS and robustness.
>
> - Although it is hard to provide theoretical aspect for PRS currently, we think that our work can give the insight for the robustness in a perspective of internal DB/DR, and PRS can contribute to the improvement of robustness in neural networks without adversarial examples.
>
>
> [1] Ilyas, Andrew, et al. "Adversarial examples are not bugs, they are features." NeurIPS 2019.
>
> [2] Madry, Aleksander, et al. "Towards deep learning models resistant to adversarial attacks." ICLR 2018.
>
> [3] Sokolić, Jure, et al. "Robust large margin deep neural networks." IEEE Transactions on Signal Processing 65.16 (2017): 4265-4280.
>
> [4] Montufar, Guido F., et al. "On the number of linear regions of deep neural networks." Advances in neural information processing systems 27 (2014).
>
> ---
>
> **Q2. Some definitions are confusing. For example, in Definition 2 (Decision Region (DR)), if the i-th layer has only one neuron, what are $D_l$ and $|$$DR_{V_{l}}$$|$ ? From the definition, it seems that $D_l$ =1 and $|$$DR_{V_{l}}$$|$=2 in this case, but this seems strange. Please give one small example for each definition to make things clear.**
>
> ---
>
> - One neuron in the internal layer can represent the half-space (i.e., the negative/positive side depending on its activation value), and we define the internal DB as a set of samples with an activation value of zero. The half-space can be used to represent the side where a given sample resides.
>
> - As $V_l$ is the indicator vector to represent the half-space, if we consider the one neuron in the $l$-th layer, it can have -1 or 1 value for the given sample $x$. So if $D_l=1$, $|V_l|=1$ and $|DR_{V_l}|=1$, because the input space is only seperated by one internal DB and it can be expressed by one value. For $D_l>1$, each neuron has each internal DB and half-space. Then, internal decision region (DR) is represented by the intersection of each half-space (see color-shaded area in the right part of Figure 1 and Appendix I).
>
> - For a more accurate understanding, we also provide an illustration for the concept of internal DB and DR in Appendix I in our revised draft.
>
> ---
> We sincerely thank you for the insightful comments. Please let us know if there is anything else we should address, or additional questions.

---

> > ### Comment · Reviewer_mBon · 2022-11-17
> > **Thanks for the reply**
> >
> > Thanks for the detailed reply from the authors. However, I still think that some theoretical analysis is needed and will help us have a better understanding of the relationship between PRS and the robustness of neural networks. Therefore, I will keep the current score.

---

### Official Review · Reviewer_x9iE · 2022-10-27

**Confidence:** 4
**Correctness:** 3
**Technical Novelty And Significance:** 2
**Empirical Novelty And Significance:** 3
**Recommendation:** 6

**Clarity, Quality, Novelty And Reproducibility:**

The empirical findings of this paper are pretty interesting to understand more about adversarial examples and robustness. Based on  the empirical findings, the paper proposes a method without adversarial training and adversarial examples to improve the robustness. Unfortunately, this seems not be able to generalize for all datasets. Additionally, the terminologies and mathematical notions used in this paper need to be revised for avoiding misleading.

**Details Of Ethics Concerns:**

There is no ethics concerns.

**Strength And Weaknesses:**

Strength
- Some empirical findings are quite interesting.
- If a model can really defend without adversarial training, it is really great. However, it seems not the case for Cifar100 as in the supplementary material.

Weaknesses
- The concept of decision boundary and decision region as defined in the paper is confusing and misleading.  Because the decision region is normally relevant one class and represents the date examples classified to this class by a deep net, while the decision boundary represent the boundaries of the decision regions. I believe it should be better if the authors use the terminology like activation patterns.
- Some behavior experiments lack of details and descriptions, hence hard to follow. For example, cosine similarity matrix of the final layer on Network A/B: cosine similarity of what and what (e.g., representations or weights). Also, Figure 5a is hard to interpret.
- Mathematical notions used is not solid. For example, the one to define $PRS(X, f, l)$: why do we need $\forall V_l \in \{-1, 1\}^D$; the one to define $MR_{l,c}$: if $DR \in PRS(X_c, f, l)$ then $DR \in X_c$, hence $DR \cap X_c = DR$.
- It is great if $L_{ham}$ and $L_{MVR}$ can help to defense for all datasets. But it seems that for Cifar100, it still needs adversarial training.

**Summary Of The Paper:**

This paper first studies the relationship of the decision regions induced by the penultimate layer of a deep net to adversarial robustness and then relies on empirical findings to propose strengthening adversarial robustness without adversarial training.

**Summary Of The Review:**

The empirical findings of this paper are pretty interesting. However, the proposed approach seems not able to generalize to all datasets because for Cifar100, it still needs adversarial training.

---

> ### Author Response · Authors · 2022-11-11
> **Response to Reviewer x9iE (2/2)**
>
> ---
>
> **Q3. Mathematical notions used is not solid. For example, the one to define $PRS(X,f,l)$ : why do we need all $V_l$ $\in$ -1, 1; the one to define $MR_{l,c}$: if $DR$ $\in$ $PRS(\mathbf{X}, f, l)$ then $DR \in X_{c}$, hence $DR \cap \mathbf{X}_c = DR$**
>
> ---
>
>
> - We are sorry for the confused mathmatical notions. To compute $PRS(\mathbf{X}, f, l)$, we need to collect the sign of feature vector for dataset $X$ at the target layer $l$. After stacking, we take a unique set over the collected sign vectors. As a result, we redefine the defintion of PRS as,
>
> $$ PRS(\mathbf{X},f,l) = \\{ DR_\mathbf{V_l}|\mathbf{V_l}=sign(f_{l:1}(x)), \forall x\in\mathbf{X} \\}. $$
>
> We note that PRS is a unique set of collected vectors, and $V_l$ is needed to define the sign of activation vector for a given train sample $x$.
>
> - We try to represent the intersection between space ($DR_\mathbf{V_l}$) and the discrete training samples $x\in\mathbf{X_c}$ in original definitions (see an illustration in Appendix I). However, we agree that the described defintion of $MR_{l,c}$ may cause the confusion. As a result, we redefine the $MR_{l,c}$ to clarify the concept.
>
>
> $$ MR_{l,c}= \underset{DR_\mathbf{V_l}\in PRS(\mathbf{X_c,f,l})}{\mathrm{argmax}} |\\{x|sign(f_{l:1}(x))=\mathbf{V_l}, x\in\mathbf{X_c}\\}|$$
>
> ---
>
>  **Q4. It is great if  and  can help to defense for all datasets. But it seems that for Cifar100, it still needs adversarial training.**
>
> ---
>
> - In CIFAR-100 experiment, although the proposed regularizers ($L_{PRS}$) cannot beat the adversarial training ($AT$), the model with $L_{PRS}$ shows the significant improvement in robust accuracy (e.g., SA: 0.37% -> 14.18%, AA: 0.21% -> 12.4%, Table 6 in the Appendix F.2) compared to the standard training. We also think that $L_{PRS}$ can have strength in a pespective of the computation cost (e.g., Time/Epoch (s)) against $AT$.
>
> - It denotes that the properties of feature embeddings described by PRS (we note that the PRS is computed in the penultimate layer) can be an alternative to understand the robustness of the network. In Table 6, we can also confirm that the $L_{PRS}+AT$ acheives higher performance by mitigating the degradation of clean accuracy than $AT$.
> In other words, using $L_{PRS}+AT$, which directly reduces PR, can be more powerful than using only $AT$. That is, we think that a method which directly reduces PR can help to improve robustness.
>
> - Although the $L_{PRS}$ is proposed as an application which utilizes our empirical study between PRS and robustness, we think that the concept of PRS cannot be limited to just the regularizer. We believe that PRS can be a starting point to provide novel insight for the robustness of the deep neural networks.
>
> ---
> We sincerely thank you for the insightful comments. Please let us know if there is anything else we should address, or additional questions.

---

> > ### Comment · Reviewer_x9iE · 2022-12-11
> > **My feedback**
> >
> > Thanks for your response to me. I decide to increase my score to 6. It would be more significant if $L_{RPS}$ alone without AT can improve the adversarial robustness.

---

> ### Author Response · Authors · 2022-11-11
> **Response to Reviewer x9iE (1/2)**
>
> We sincerely appreciate your efforts and constructive comments to review our manuscript. We have revised the manuscript by faithfully reflecting your comments. We respond to your comments in the following.
>
> ---
>
> **Q1. The concept of decision boundary and decision region as defined in the paper is confusing and misleading. Because the decision region is normally relevant one class and represents the date examples classified to this class by a deep net, while the decision boundary represent the boundaries of the decision regions. I believe it should be better if the authors use the terminology like activation patterns.**
>
> ---
>
> - As you mentioned, the decision boundary (DB) and region (DR) are generally defined for the class logits of the network. In particular, a set of samples with the same logits between two classes is defined as the DB [1]. From the concept of DB, we can consider the simliar concept related to the activation of the internal features (i.e., zero value of the activation) to investigate the characteristics of the network. We can also find the similar internal boundary concept in the previous work [2].
>
> - As a result, we expand the DB and DR to the internal layer of the network (to alleviate confusion, we used the notion of 'internal' in the main paper). We believe that the input space can be divided by the internal DBs, and the internal DBs can be represented by the activation on/off which comprise the internal DRs.
>
> - To clarify the terminologies, we have revised the name of DB and DR that the 'internal' concept can be reflected. We also provide an illustration for the concept of internal DB and DR in Appendix I in our revised draft.
>
> [1] Fawzi., et al. "Empirical study of the topology and geometry of deep networks." CVPR 2018.
>
> [2] Jeon., et al. "An efficient explorative sampling considering the generative boundaries of deep generative neural networks." AAAI 2020.
>
> ---
>
> **Q2. Some behavior experiments lack of details and descriptions, hence hard to follow. For example, cosine similarity matrix of the final layer on Network A/B: cosine similarity of what and what (e.g., representations or weights). Also, Figure 5a is hard to interpret.**
>
> ---
>
> - Sorry for the lack of details for the performed experiments. We are modifying the manuscript to improve the clarity of the procedure of the experiments. To visualize the cosine similarity matrix for the weight parameters in the final layer on Network A/B, we consider the weight parameters $W_{c_i} \in \mathbb{R}^{D_{L-1}\times 1}$ for each class $c_i$ where $D_{L-1}$ denotes the dimension of output on the penultimate layer. The cosine similarity $CS(\cdot, \cdot)$ between two classes $c_i$ and $c_j$ is defined as,
> $$ CS(W_{c_i}, W_{c_j})=\frac{W_{c_i}^TW_{c_j}}{|W_{c_i}||W_{c_j}|}. $$
>
> - When we consider the output on the penultimate layer as the embedding of the network, the weight parameter $W_{c_i}$ is the normal vector for the linear DB of the class $c_i$. As a result, the defined cosine similarity can measure the alignment among the normal vectors from two classes. The Figure 5 (b) denotes $10\times10$ matrix which include the cosine similarity between each class.
>
> - In Figure 5 (a), we try to visualize the change of logits for the adversarial examples in each network (We note that network A and B are have different PRS). For the $\textit{horse}$ example, when we perform the adversarial attack, the index of the original class ($c_7$) is not changed on Network B while the index is changed on Network A (see the highest index for the black/orange line).
>
> - We conjecture that this phenomenon can be caused the weight parameters on the final layer. To verify our hypothesis, we measure and visualize the cosine similarity matrix as we discussed before. We have revised the Figure 5 (a) to improve the interpretation (e.g., Adding horizontal lines for the highest logit value for each logit vector).

---

### Author Response · Authors · 2022-11-15
**General Response**

Dear Reviewers and AC,

We really appreciate all the reviewers for their constructive comments. We have responded to the common comments as well as individual comments from the reviewers below, and believe that we have successfully responded to all of them. Here we briefly summarize the updates we have made to the revised version of the paper:

- We provided an illustration for the terminologies of internal DB, DR and PRS in Appendix I (Figure 21 and 22).
- We provided a toy example for PRS regularizer to support our claims in Appendix J (Figure 23).
- We revised the mathematical notions (Deinifition 3 and 4).
- We revised the terms (e.g., PRS ratio and PRS inclusion ratio) in Section 1.
- We inserted dashed horizontal lines to improve the interpretation (Figure 5-(a)).
- We inserted a description of the AA experimental results for the standardly trained model in Appendix F.1.

We sincerely believe that these updates may help us better deliver the benefits of the our work to the ICLR community.

Thank you very much,

Authors.

---

### Decision · Program_Chairs · 2023-01-20

**Decision:**

Reject

**Justification For Why Not Higher Score:**

The proposed method does not result in a sufficient improvement in robustness w.r.t. diverse attacks. Adversarial training is still required, e.g. for Cifar 100.

**Justification For Why Not Lower Score:**

N/A

**Metareview: Summary, Strengths And Weaknesses:**

This paper studies the relationship of the decision regions in the penultimate layer deep NNs and deduces an approach to increase the robustness. The introduced concept uses the Populated Region Set (PRS) to characterize the complexity of DNNs to improve adversarial robustness against PGD.
While the paper addresses in important topic and proposes an interesting analysis, the remaining criticism mostly focuses on the experimental evaluation - showing relatively strong improvements w.r.t. PGD but not against AA. As a result, it remains unclear whether the decision regions are sufficiently explored.